Accepted at the ICLR 2024 Workshop on AI4Differential Equations In Science

# LEARNING TIME-DEPENDENT PDE VIA GRAPH NEURAL NETWORKS AND DEEP OPERATOR NETWORK FOR ROBUST ACCURACY ON IRREGULAR GRIDS

**Sung Woong Cho**[*]
Stochastic Analysis and Application Research Center
Korea Advanced Institute of Science and Technology
Deajeon, 34141, Republic of Korea
swcho95kr@kaist.ac.kr

**Jae Yong Lee**[*]
Department of Artificial Intelligence
Chung-Ang University
Seoul, 06974, Republic of Korea
jaeyong@cau.ac.kr

**Hyung Ju Hwang** [†]
Department of Mathematics
Pohang University of Science and Technology
Pohang, 37673, Republic of Korea
hjhwang@postech.ac.kr

## ABSTRACT

There has been growing interest in models that learn the operator from the parameters of a partial differential equation (PDE) to the corresponding solutions. Deep Operator Network (DeepONet) and Fourier Neural operator, among other models, have been designed with structures suitable for handling functions as inputs and outputs, enabling real-time predictions as surrogate models for solution operators. There has also been significant progress in the research on surrogate models based on graph neural networks (GNNs), specifically targeting the dynamics in time-dependent PDEs. In this paper, we propose GraphDeepONet, an autoregressive model based on GNNs, to effectively adapt DeepONet, which is well-known for successful operator learning. GraphDeepONet exhibits robust accuracy in predicting solutions compared to existing GNN-based PDE solver models. It maintains consistent performance even on irregular grids, leveraging the advantages inherited from DeepONet and enabling predictions on arbitrary grids. Additionally, unlike traditional DeepONet and its variants, GraphDeepONet enables time extrapolation for time-dependent PDE solutions.

## 1 INTRODUCTION

In recent years, operator learning frameworks have gained significant attention in the field of artificial intelligence. The primary goal of operator learning is to employ neural networks to learn the mapping from the parameters (external force, initial, and boundary condition) of a PDE to its corresponding solution operator. To accomplish this, researchers are exploring diverse models and methods, such as the deep operator network (DeepONet) (Lu et al., 2019) and Fourier neural operator (FNO) (Li et al., 2020), to effectively handle functions as inputs and outputs of neural networks. These frameworks present promising approaches to solving PDEs by directly learning the underlying operators from available data. Several studies (Lu et al., 2022; Goswami et al., 2022) have conducted comparisons between DeepONet and FNO, and with theoretical analyses (Lanthaler et al., 2022; Kovachki et al., 2021a) have been performed to understand their universality and approximation bounds.

In the field of operator learning, there is an active research focus on predicting time-evolving physical quantities. The DeepONet can be applied to simulate time-dependent PDEs by incorporating a time variable, denoted as $t$, as an additional input with spatial variables, denoted as $x$. However,

---

[*]These authors contributed equally.
[†]corresponding author

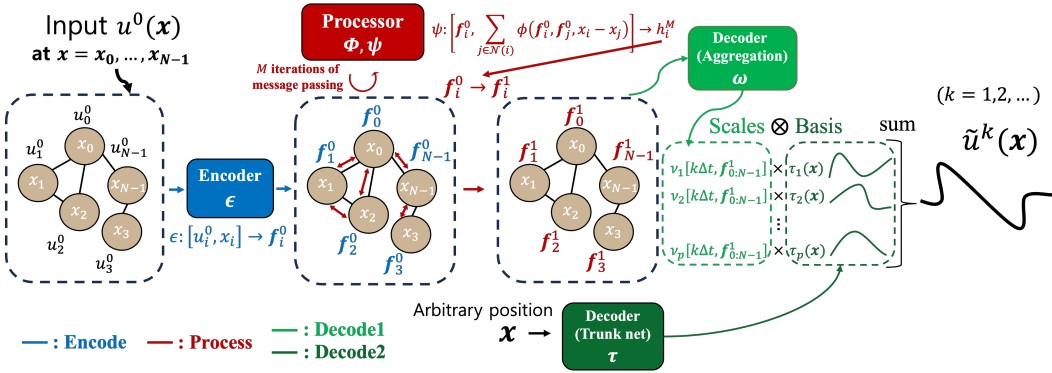

Figure 1: Framework of the proposed GraphDeepONet

the use of both $t$ and $x$ as inputs at once to the DeepONet can only predict solutions within a fixed time domain and they should be treated differently from a coefficient and basis perspective. FNO (Li et al., 2020; Kovachki et al., 2021b) also introduces two methods specifically designed for this purpose: FNO-2d, which utilizes an autoregressive model, and FNO-3d. However, a drawback of FNO is its reliance on a fixed uniform grid. To address this concern, recent studies have explored the modified FNO (Lingsch et al., 2023; Lin et al., 2022), such as geo-FNO (Li et al., 2022) and F-FNO (Tran et al., 2023).

To overcome this limitation, researchers have explored the application of GNNs and message passing methods (Scarselli et al., 2008; Battaglia et al., 2018; Gilmer et al., 2017; Sanchez-Gonzalez et al., 2020; Pfaff et al., 2021; Lienen & Günnemann, 2022) to learn time-dependent PDE solutions. In particular, Brandstetter et al. (2022) and Boussif et al. (2022) focused on solving the time-dependent PDE based on GNNs. Brandstetter et al. (2022) proposed a Message-Passing Neural PDE Solver (MP-PDE) that utilizes message passing to enable the learning of the solution operator for PDEs, even on irregular domains. However, a limitation of their approach is that it can only predict the solution operator on the same irregular grid used as input, which poses challenges for practical simulation applications. To address this limitation, Boussif et al. (2022) introduced the Mesh Agnostic Neural PDE solver (MAgNet), which employs a network for interpolation in the feature space. This approach allows for more versatile predictions and overcomes the constraints of using the same irregular grid for both input and solution operator prediction. We aim to employ the DeepONet model, which learns the basis of the target function's spatial domain, to directly acquire the continuous space solution operator of time-dependent PDEs without requiring additional interpolation steps. By doing so, we seek to achieve more accurate predictions at all spatial positions without relying on separate interpolation processes. Our main contributions can be summarized as follows:

- By effectively incorporating time information into the branch net using a GNN, GraphDeepONet enables time extrapolation prediction for PDE solutions, a task that is challenging for traditional DeepONet and its variants.

- Our method exhibits robust accuracy in predicting the solution operator at arbitrary positions of the input on irregular grids compared to other graph-based PDE solver approaches. The solution obtained through GraphDeepONet is a continuous solution in the spatial domain.

## 2 GRAPHDEEPONET FOR TIME-DEPENDENT PDES

For a fixed set of positional sensors $x_i$ ($0 \le i \le N - 1$), we formulate a graph $G = (\mathcal{V}, \mathcal{E})$, where each node $i$ belongs to $\mathcal{V}$ and each edge $(i, j)$ to $\mathcal{E}$. The nodes represent grid cells, and the edges signify local neighborhoods.

**Encoder $\epsilon$.** The encoder maps node embeddings from the function space to the latent space. For a given node $i$, it maps the last solution values at node position $\boldsymbol{x}_i$, denoted as $u_i^0 := u^0(\boldsymbol{x}_i)$, to the latent embedding vector. Formally, the encoding function $\epsilon : \mathbb{R}^{1+d} \to \mathbb{R}^{d_{\text{lat}}}$ produces the node embedding vector $\boldsymbol{f}_i^0$ as follows:

$$\boldsymbol{f}_i^0 := \epsilon\left(u_i^0, \boldsymbol{x}_i\right) \in \mathbb{R}^{d_{\text{lat}}}, \tag{1}$$

where $\epsilon$ is multilayer perceptron (MLP). It is noteworthy that the sampling method, which includes both the number of sensors $N$ and their respective locations $\boldsymbol{x}_i$ for $0 \leq i \leq N-1$, can differ for each input.

**Processor $\phi, \psi$.** The processor approximates the dynamic solution of PDEs by performing $M$ iterations of learned message passing, yielding intermediate graph representations. The update equations are given by

$$\boldsymbol{m}_{ij}^m = \phi(\boldsymbol{h}_i^m, \boldsymbol{h}_j^m, \boldsymbol{x}_i - \boldsymbol{x}_j), \tag{2}$$

$$\boldsymbol{h}_i^{m+1} = \psi\left(\boldsymbol{h}_i^m, \sum_{j \in \mathcal{N}(i)} \boldsymbol{m}_{ij}^m\right), \tag{3}$$

for $m = 0, 1, ..., M-1$ with $\boldsymbol{h}_i^0 = \boldsymbol{f}_i^0$, where $\mathcal{N}(i)$ denotes the neighboring nodes of node $i$. Both $\phi$ and $\psi$ are implemented as MLPs. The use of relative positions, i.e., $\boldsymbol{x}_j - \boldsymbol{x}_i$, capitalizes on the translational symmetry inherent in the considered PDEs. After the $M$ iterations of message passing, the processor emits a vector $\boldsymbol{h}_i^M$ for each node $i$. This is used to update the latent vector $\boldsymbol{f}_i^0$ as follows:

$$\boldsymbol{f}_i^1 = \boldsymbol{f}_i^0 + \boldsymbol{h}_i^M, \quad 0 \leq i \leq N-1. \tag{4}$$

The updated latent vector $\boldsymbol{f}_{0:N-1}^1 := \{\boldsymbol{f}_i^1\}_{i=0}^{N-1}$ is used to predict the next time step solution $u^1(\boldsymbol{x})$.

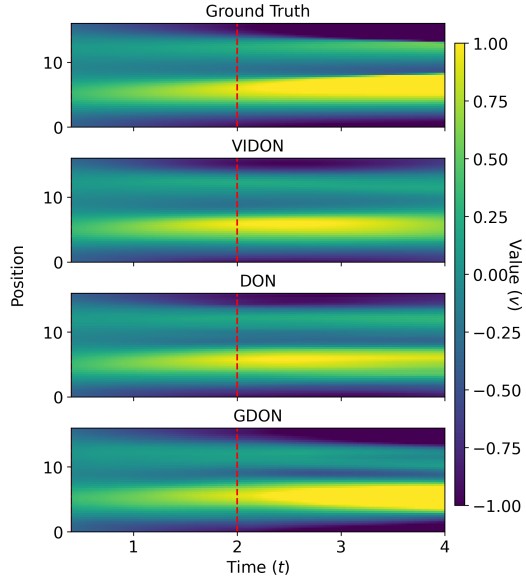

Figure 2: Solution profile in Burgers' equation for time extrapolation simulation using DeepONet, VIDON, and GraphDeepONet.

**Decoder1 - Soft attention aggregation $\omega$.**
We first predict the $p-$coefficients for each next timestep. Here, we use the soft attention aggregation layer with the feature-level gating described by Li et al. (2019). The soft attention aggregation $\boldsymbol{\nu} : \mathbb{R}^{d_{\text{lat}} \times N} \to \mathbb{R}^p$ consists of two neural networks to calculate the attention scores and latent vectors as follows:

$$\boldsymbol{\nu}[\boldsymbol{f}_{0:N-1}^1, \Delta t] := \sum_{i=0}^{N-1} \overbrace{\frac{\exp\left(\omega_{\text{gate}}(\boldsymbol{x}_i, \boldsymbol{f}_i^1)/\sqrt{d_{\text{lat}}}\right)}{\sum_{j=0}^{N-1} \exp\left(\omega_{\text{gate}}(\boldsymbol{x}_j, \boldsymbol{f}_j^1)/\sqrt{d_{\text{lat}}}\right)}}^{\text{attention score}} \odot \omega_{\text{feature}}(\Delta t, \boldsymbol{f}_i^1), \tag{5}$$

where $\odot$ represents the element-wise product, and $\omega_{\text{gate}} : \mathbb{R}^{d_{\text{lat}}+d} \to \mathbb{R}^p$ and $\omega_{\text{feature}} : \mathbb{R}^{d_{\text{lat}}+1} \to \mathbb{R}^p$ are MLPs. Note that $\boldsymbol{\nu}$ is well-defined for any number of sensors $N \in \mathbb{N}$.

**Decoder2 - Inner product of coefficients and basis $\tau$.** The final output is reconstructed using the $p-$coefficients $\boldsymbol{\nu}[\boldsymbol{f}_{0:N-1}^1, \Delta t]$ and trained global basis via trunk net $\boldsymbol{\tau}(\boldsymbol{x}) = [\tau_1(\boldsymbol{x}), ..., \tau_p(\boldsymbol{x})]$ with $\tau_j : \mathbb{R}^d \to \mathbb{R}$. The next timestep is predicted as

$$\widetilde{u}^1(\boldsymbol{x}) = \sum_{j=1}^{p} \nu_j[\boldsymbol{f}_{0:N-1}^1, \Delta t]\tau_j(\boldsymbol{x}), \tag{6}$$

where $\boldsymbol{\nu}[\boldsymbol{f}_{0:N-1}^1, \Delta t] := [\nu_1, \nu_2, ..., \nu_p] \in \mathbb{R}^p$. The GraphDeepONet is trained using the mean square error $\text{Loss}^{(1)} = \text{MSE}(\widetilde{u}^1(\boldsymbol{x}), u^1(\boldsymbol{x}))$. Since the GraphDeepONet use the trunk net to learn the global basis, it offers a significant advantage in enforcing the boundary condition $\mathcal{B}[u] = 0$ as hard constraints. The GraphDeepONet can enforce periodic boundaries, unlike other graph-based methods, which often struggle to ensure such precise boundary conditions (See Appendix B.6).

Table 1: Mean Rel. $L^2$ test errors with standard deviations for 3 types of Burgers' equation dataset using regular/irregular sensor points. Three training trials are performed independently.

| Type of sensor points | Data | FNO-based model | | DeepONet variants | | Graph-based model | | GraphDeepONet (Ours) |
|---|---|---|---|---|---|---|---|---|
| | | FNO-2D | F-FNO | DeepONet | VIDON | MP-PDE | MAgNet | |
| Regular | E1 | $0.1437\pm 0.0109$ | $\mathbf{0.1060}\pm 0.0021$ | $0.3712\pm 0.0094$ | $0.3471\pm 0.0221$ | $0.3598\pm 0.0019$ | $0.2399\pm 0.0623$ | $0.1574\pm 0.0104$ |
| | E2 | $0.1343\pm 0.0108$ | $\mathbf{0.1239}\pm 0.0025$ | $0.3688\pm 0.0204$ | $0.3067\pm 0.0520$ | $0.2622\pm 0.0019$ | $0.2348\pm 0.0153$ | $0.1716\pm 0.0350$ |
| | E3 | $0.1551\pm 0.0014$ | $\mathbf{0.1449}\pm 0.0053$ | $0.2983\pm 0.0050$ | $0.2691\pm 0.0145$ | $0.3548\pm 0.0171$ | $0.2723\pm 0.0628$ | $0.2199\pm 0.0069$ |
| Irregular | E1 | - | $0.3793\pm 0.0056$ | $0.3564\pm 0.0467$ | $0.3430\pm 0.0492$ | $0.2182\pm 0.0108$ | $0.4106\pm 0.0864$ | $\mathbf{0.1641}\pm 0.0006$ |

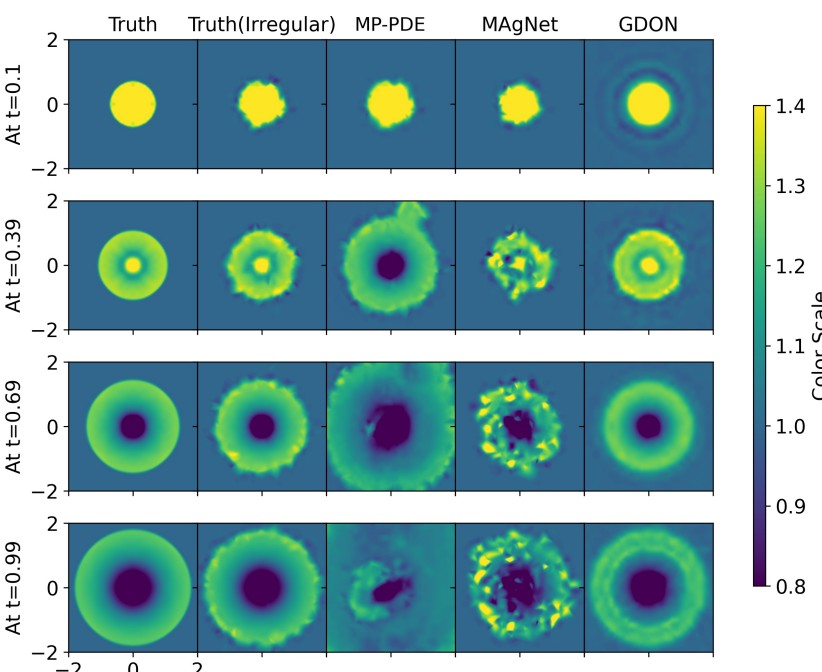

Figure 3: Prediction of 2D shallow water equations on irregular sensor points with distinct training sensor points using graph-based models and GraphDeepONet. The Truth (irregular), MP-PDE, and MAgNet plot the solutions through interpolation using values from the irregular sensor points used during training, whereas GraphDeepONet predicts solutions for all grids directly.

## 3 EXPERIMENTS

We conduct experiments comparing the proposed GraphDeepONet model with other benchmark models. Firstly, we explore the simulation of time-dependent PDEs by comparing the original DeepONet and VIDON with GraphDeepONet for regular and irregular sensor points. Specifically, we assess how well GraphDeepONet predicts in arbitrary positions, especially concerning irregular sensor points, compared to models such as MP-PDE, and MAgNet. Furthermore, we include FNO-2D, a well-established model known for operator learning, in our benchmark comparisons. Given the difficulty FNO faces in handling input functions with irregular sensor points, we also consider Factorized FNO (F-FNO) (Tran et al., 2023), which extends FNO to irregular grids (See Appendix B.3). We consider the 1D Burgers' equation data from Brandstetter et al. (2022), the 2D shallow water equation data from Takamoto et al. (2022), and the 2D Navier-Stokes (N-S) equation data from Kovachki et al. (2021b). For datasets with periodic boundaries, the GraphDeepONet leveraged the advantage of enforcing the condition (See Appendix B.6). The PyTorch Geometric library (Fey & Lenssen, 2019) is used for all experiments. The relative $L^2$ error by averaging the prediction solutions for all time is used for error estimate. See Appendix B for more details.

**Comparison with DeepONet and its variants** The fourth and fifth columns in Table 1 display the training results for DeepONet and VIDON, respectively. The DeepONet and VIDON struggled to accurately predict the solutions of Burgers's equation. This is because DeepONet and VIDON lack

Table 2: Mean Rel. $L^2$ test errors for 2D shallow water equation data using regular/irregular sensor points.

| Data | Type of data | FNO-2D | F-FNO | MP-PDE | MAgNet | GraphDeepONet (Ours) |
|------|--------------|--------|-------|--------|--------|----------------------|
| 2D shallow | Regular | $0.0051_{\pm0.0024}$ | $0.0033_{\pm0.0014}$ | $\mathbf{0.0015}_{\pm0.0006}$ | $0.0073_{\pm0.0014}$ | $0.0094_{\pm0.0027}$ |
| | Irregular I | - | $0.0503_{\pm0.0041}$ | $0.1693_{\pm0.0338}$ | $0.0949_{\pm0.0635}$ | $\mathbf{0.0137}_{\pm0.0078}$ |
| | Irregular II | - | $0.0494_{\pm0.0012}$ | $0.1698_{\pm0.0395}$ | $0.0917_{\pm0.0630}$ | $\mathbf{0.0148}_{\pm0.0121}$ |
| | Irregular III | - | $0.0478_{\pm0.0018}$ | $0.0982_{\pm0.0729}$ | $0.0709_{\pm0.0184}$ | $\mathbf{0.0140}_{\pm0.0086}$ |
| 2D N-S | Regular | $\mathbf{0.0351}_{\pm0.0132}$ | $0.0323_{\pm0.0015}$ | $0.4940_{\pm0.0185}$ | $0.3761_{\pm0.0010}$ | $0.1323_{\pm0.0114}$ |
| | Irregular I | - | $0.2055_{\pm0.0251}$ | $0.7817_{\pm0.2909}$ | $0.4139_{\pm0.0584}$ | $\mathbf{0.1223}_{\pm0.0020}$ |
| | Irregular II | - | $0.2426_{\pm0.1311}$ | $\mathbf{0.1163}_{\pm0.0053}$ | $0.4142_{\pm0.0462}$ | $0.1271_{\pm0.0022}$ |
| | Irregular III | - | $0.3030_{\pm0.0813}$ | $\mathbf{0.1240}_{\pm0.0037}$ | $0.3982_{\pm0.0283}$ | $0.1279_{\pm0.0056}$ |

universal methods to simultaneously handle input and output at multiple timesteps. Figure 2 compares the time extrapolation capabilities of existing DeepONet models. To observe extrapolation, we trained our models using data from time $T_{\text{train}} = [0, 2]$, with inputs ranging from 0 to 0.4, allowing them to predict values from 0.4 to 2. Subsequently, we evaluated the performance of DeepONet, VIDON, and our GraphDeepONet by predicting data $T_{\text{extra}} = [2, 4]$, a range on which they had not been previously trained. Our model clearly demonstrates superior prediction performance when compared to VIDON and DeepONet. In contrast to DeepONet and VIDON, which tend to maintain the solutions within the previously learned domain $T_{\text{train}}$, the GraphDeepONet effectively learns the variations in the PDE solutions over time, making it more proficient in predicting outcomes for time extrapolation.

**Comparison with GNN-based PDE-solvers** The third, sixth, and seventh columns of Table 1 depict the accuracy of the FNO-2D and GNN-based models. While FNO outperformed the other models on a regular grid, unlike graph-based methods and our approach, it is not applicable to irregular sensor points, which is specifically designed for uniform grids. F-FNO also faces challenges when applied to the irregular grid. When compared to GNN-based models, with the exception of F-FNO, our model slightly outperformed MP-PDE and MAgNet, even on an irregular grid. Table 2 summarizes the results of our model along with other models, when applied to various irregular grids for 2D shallow water equation and 2D N-S equation, namely, Irregular I,II, and III for each equation. Remarkably, on one specific grid, MP-PDE outperformed our model. However, the MP-PDE has a significant inconsistency in the predicted performance. In contrast, our model consistently demonstrated high predictive accuracy across all grid cases. This is because, unlike other methods, the solution obtained through GraphDeepONet is continuous in the spatial domain. Figure 3 displays the time-evolution predictions of models trained on the shallow water equation for an initial condition. The GNN-based models are trained on fixed irregular sensors as seen in the second column and are only capable of predicting on the same grid, necessitating interpolation for prediction. In contrast, GraphDeepONet leverages the trunk net, enabling predictions at arbitrary grids, resulting in more accurate predictions.

## 4 CONCLUSION AND DISCUSSION

The proposed GraphDeepONet represents a significant advancement in the realm of PDE solution prediction. Its unique incorporation of time information through a GNN in the branch net allows for precise time extrapolation, a task that has long challenged traditional DeepONet and its variants. Additionally, our method outperforms other graph-based PDE solvers, particularly on irregular grids, providing continuous spatial solutions. Furthermore, GraphDeepONet offers theoretical assurance, demonstrating its universal capability to approximate continuous operators across arbitrary time intervals. Altogether, these innovations position GraphDeepONet as a powerful and versatile tool for solving PDEs, especially in scenarios involving irregular grids. While our GraphDeepONet model has demonstrated promising results, one notable limitation is its current performance on regular grids, where it is outperformed by FNO. Addressing this performance gap on regular grids remains an area for future improvement. As we have employed the temporal bundling method in our approach, one of our future endeavors includes exploring other techniques utilized in DeepONet-related models and GNN-based PDE solver models to incorporate them into our model. Furthermore, exploring the extension of GraphDeepONet to handle more complex 2D time-dependent PDEs or the Navier-Stokes equations, could provide valuable insights for future works and applications.

ACKNOWLEDGMENTS

Jae Yong Lee was supported by Institute for Information & Communications Technology Planning & Evaluation (IITP) through the Korea government (MSIT) under Grant No. 2021-0-01341 (Artificial Intelligence Graduate School Program (Chung-Ang University)). Hyung Ju Hwang was supported by the National Research Foundation of Korea(NRF) grant funded by the Korea government(MSIT) (No. RS-2023-00219980 and RS-2022-00165268) and by Institute for Information & Communications Technology Promotion (IITP) grant funded by the Korea government(MSIP) (No.2019-0-01906, Artificial Intelligence Graduate School Program (POSTECH)).

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

# A    NOTATIONS

The notations in the paper is summarized in Table 3.

Table 3: Notations

| Notation | Meaning |
|---|---|
| $t$ | the spatial variable |
| $d$ | the dimension of spatial domain |
| $\boldsymbol{x}$ | the spatial variable in $d$ dimension |
| $\boldsymbol{x}_i \ (i = 0, 1, ..., N-1)$ | the $N$-fixed sensor point in the spatial domain |
| $\Delta t$ | the discretized time |
| $K_{\text{frame}} + 1$ | the number of frames in one solution trajectory |
| $K$ | the number of grouping frames for temporal bundling method |
| $u^k(\boldsymbol{x}) \ (k = 0, 1, ..., K_{\text{frame}})$ | the solution at time $t = k\Delta t$ |
| $\bar{u}^k(\boldsymbol{x}) \ (k = 0, 1, ..., K_{\text{frame}})$ | the values of solution at time $t = k\Delta t$ in fixed sensor points |
| $\widetilde{u}^k(\boldsymbol{x}) \ (k = 0, 1, ..., K_{\text{frame}})$ | the approximated solution at time $t = k\Delta t$ |
| $\mathcal{G}^{(k)}$ | the operator from the initial condition to the solution at time $k\Delta t$ |
| $\mathcal{G}_{\text{GDON}}$ | the approximated operator using GraphDeepONet |
| $\mathcal{G}_{\text{graph}}$ | the approximated operator using other graph-based PDE solver |
| $p$ | the number of basis (or coefficients) in DeepONet |
| $\boldsymbol{\nu}$ | the branch net (or decoder) in DeepONet (or GraphDeepONet) |
| $\boldsymbol{\tau}$ | the trunk net in DeepONet (or GraphDeepONet) |
| $\boldsymbol{\epsilon}$ | the encoder in GraphDeepONet |
| $\boldsymbol{\phi}, \boldsymbol{\psi}$ | the neural networks of processor in GraphDeepONet |
| $\boldsymbol{\omega}$ | the neural network of decoder in GraphDeepONet |
| $\boldsymbol{f}_i \ (i = 0, 1, ..., N-1)$ | the feature vector at node $i$ |

# B    DETAILS ON EXPERIMENTS AND ADDITIONAL EXPERIMENTS

## B.1    DETAIL SETTING ON GRAPH

The edges $(i, j) \in \mathcal{E}$ are constructed based on the proximity of node positions, connecting nodes within a specified distance. In actual experiments, we considered nodes as grids with given initial conditions. There are broadly two methods for defining edges. One approach involves setting a threshold based on the distances between grids in the domain, connecting edges if the distance between these grids is either greater or smaller than the specified threshold value. Another method involves utilizing classification techniques, such as the $k$-nearest neighbors ($k$-NN) algorithm, to determine whether to establish an edge connection. We determined whether to connect edges based on the $k$-NN algorithm with $k =$ 6 for 1D, $k = 8$ for 2D. Therefore, the processing of $\phi$ and $\psi$ takes place based on these edges. The crucial point here is that once the Graph $G = (\mathcal{V}, \mathcal{E})$ is constructed according to a predetermined criterion, even with a different set of sensor points, $\phi$ and $\psi$ remain unchanged as processor networks applied to the respective nodes and their connecting edges.

## B.2    DATASET

Similar to other graph-based PDE solver studies (Brandstetter et al., 2022; Boussif et al., 2022), we consider the 1D Burgers' equation as

$$\partial_t u + \partial_x (\alpha u^2 - \beta \partial_x u + \gamma \partial_{xx} u) = \delta(t, x), \quad t \in T = [0, 4], x \in \Omega = [0, 16], \tag{7}$$
$$u(0, x) = \delta(0, x), \quad x \in \Omega,$$

where $\delta(t, x)$ is randomly generated as

$$\delta(t, x) = \sum_{j=1}^{5} A_j \sin(a_j t + b_j x + \phi_j) \tag{8}$$

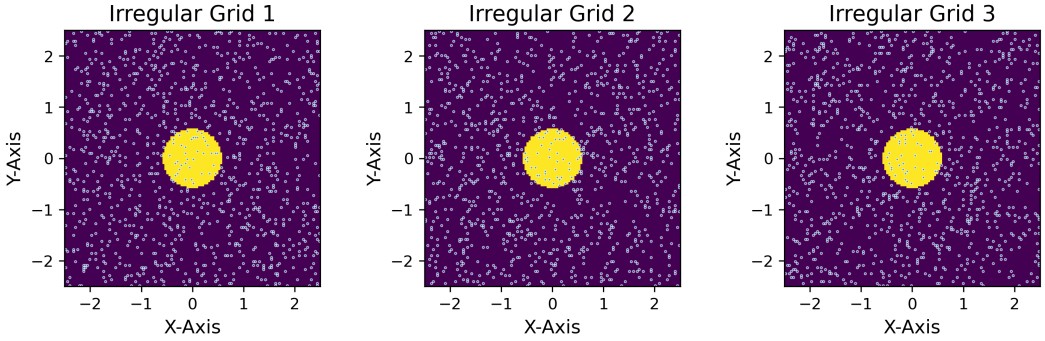

Figure 4: Three types of irregular grid (Irregular I, Irregular II, and Irregular III) used to train the models in shallow water eqaution

where $a_j$, $b_j$ and $\phi_j$ are uniformly sampled as

$$A_j \in \left[-\frac{1}{2}, \frac{1}{2}\right], a_j \in \left[-\frac{2}{5}, \frac{2}{5}\right], b_j \in \left\{\frac{\pi}{8}, \frac{2\pi}{8}, \frac{3\pi}{8}\right\}, \phi_j \in [0, 2\pi]. \tag{9}$$

We conducted a direct comparison with the models using the data E1, E2, and E3 as provided in Brandstetter et al. (2022); Boussif et al. (2022). For a more detailed understanding of the data, refer to those studies.

Also, we take the 2D shallow water equation data from Takamoto et al. (2022). The shallow water equations, which stem from the general Navier-Stokes equations, provide a suitable framework for the modeling of free-surface flow problems. In two dimensions, it can be expressed as the following system of hyperbolic PDEs

$$\frac{\partial h}{\partial t} + \frac{\partial}{\partial x}(hu) + \frac{\partial}{\partial y}(hv) = 0,$$

$$\frac{\partial(hu)}{\partial t} + \frac{\partial}{\partial x}\left(u^2 h + \frac{1}{2}gh^2\right) + \frac{\partial}{\partial y}(huv) = 0, \quad t \in [0, 1], \boldsymbol{x} = (x, y) \in \Omega = [-2.5, 2.5]^2$$

$$\frac{\partial(hv)}{\partial t} + \frac{\partial}{\partial y}\left(v^2 h + \frac{1}{2}gh^2\right) + \frac{\partial}{\partial x}(huv) = 0, \tag{10}$$

$$h(0, x, y) = h_0(x, y),$$

where $h(t, x, y)$ is the height of water with horizontal and vertical velocity $(u, v)$ and $g$ is the gravitational acceleration. We generate the random samples of initial conditions similar to the setting of Takamoto et al. (2022). The initial condition is generated by

$$h_0(x, y) = \begin{cases} 2.0, & \text{for } r < \sqrt{x^2 + y^2} \\ 1.0, & \text{for } r \geq \sqrt{x^2 + y^2} \end{cases} \tag{11}$$

where the radius $r$ is uniformly sampled from $[0.3, 0.7]$.

We utilize the same Navier-Stokes data employed in Li et al. (2020). The dynamics of a viscous fluid are described by the Navier-Stokes equation. In the vorticity formulation, the incompressible Navier-Stokes equation on the unit torus can be represented as follows:

$$\begin{cases} \frac{\partial w}{\partial t} + u \cdot \nabla w - \nu \Delta w = f, & (t, \boldsymbol{x}) \in [0, T] \times (0, 1)^2, \\ \nabla \cdot u = 0, & (t, \boldsymbol{x}) \in [0, T] \times (0, 1)^2, \\ w(0, \boldsymbol{x}) = w_0(\boldsymbol{x}), & \boldsymbol{x} \in (0, 1)^2, \end{cases} \tag{12}$$

Here, $w$, $u$, $\nu$, and $f$ represent the vorticity, velocity field, viscosity, and external force, respectively.

Table 4: The training time and inference time for the N-S equation data using GNN based models.

| Data | Model | Training time per epoch (s) | Inference time per timestep (ms) |
|---|---|---|---|
| Navier Stokes (2D Irregular I) | GDON(Ours) | 7.757 | 19.49 |
| | MAgNet | 18.09 | 48.81 |
| | MP-PDE | 4.88 | 10.1 |

### B.3 COMPARISON WITH F-FNO

Our focus is on comparing our model with existing graph neural network(GNN)-based models capable of simulating time-dependent PDEs on irregular domains, such as MP-PDE and MAgNet. Consequently, instead of considering variations of FNO, we concentrated on GNN-based PDE solvers for experiment baseline. Therefore, FNO, being a fundamental model in operator learning area, was compared only on regular grids. We included experiments comparing our model with F-FNO proposed in Tran et al. (2023), which is state-of-the-art on regular grids and applicable to irregular grids. As shown in Table 2, the F-FNO is applicable to irregular grids data, but it generally exhibits higher errors compared to GraphDeepONet. This is attributed to the limited capacity for the number of input features. We used the F-FNO model, which is built for Point Cloud data, to predict how solutions will evolve over time. At first, the model was designed to process dozens of input features, which made it difficult to include all the initial values from a two-dimensional grid.

2

### B.4 COMPUTATIONAL TIME COMPARISON WITH BENCHMARK MODELS

One significant advantage of models based on Graph Neural Networks (GNNs), such as MP-PDE, MAgNet, and GraphDeepONet (ours), compared to traditional numerical methods for solving time-dependent PDEs, lies in their efficiency during inference. In traditional numerical methods, solving PDEs for different initial conditions requires recalculating the entire PDE, and in real-time weather prediction scenarios (Kurth et al., 2022), where numerous PDEs with different initial conditions must be solved simultaneously, this can result in a substantial computational burden. On the other hand, models based on GNNs (MP-PDE, MAgNet, GraphDeepONet), including the process of learning the operator, require data for a few frames of PDE. However, after training, they enable rapid inference, allowing real-time PDE solving. More details on advantage using operator learning model compared to traiditional numerical method is explained in many studies (Goswami et al., 2022; Kovachki et al., 2021b).

Table 4 presents a computational time comparison between our proposed GraphDeepONet and other GNN-based models. Due to its incorporation of global interaction using equation 5 for a better understanding of irregular grids, GraphDeepONet takes longer during both training and inference compared to MP-PDE. However, the MAgNet model, which requires separate interpolation for irregular grids, takes even more time than MP-PDE and GraphDeepONet. This illustrates that our GraphDeepONet model exhibits a trade-off, demonstrating a stable accuracy for irregular grids compared to MP-PDE, while requiring less time than MAgNet.

### B.5 MODEL HYPERPARAMETERS FOR BENCHMARK MODELS AND OUR MODEL

We trained various models, including DeepONet and VIDON, following the architecture and sizes as well as the training hyperparameters outlined in Prasthofer et al. (2022). Additionally, MP-PDE and MAgNet utilized parameter settings as provided in Boussif et al. (2022) without modification. We trained our model, the GraphDeepONet, using the Adam optimizer, starting with an initial learning rate of 0.0005. This learning rate is reduced by 20

In the small architecture, the encoder was set up with a width of 128 and a depth of 2 for epsilon. The processor components, $\phi$, and $\psi$, each had a width of 128 and a depth of 2. We employed distinct $\phi$ and $\psi$ for each of the three message-passing steps. In the decoder, we assigned $\omega_{\text{gate}}$ and $\omega_{\text{feature}}$ for aggregation to the neural network, which had a width of 128 and a depth of 3. The trunk net, $\tau$, was configured with a width of 128 and a depth of 3.

For the large architecture, the width of all neural networks was set to 128, and the depth was set to 3, except for the trunk net. The trunk net's depth was set to 5. The number of message-passing steps was set to 3. For more specific details, refer to the code.

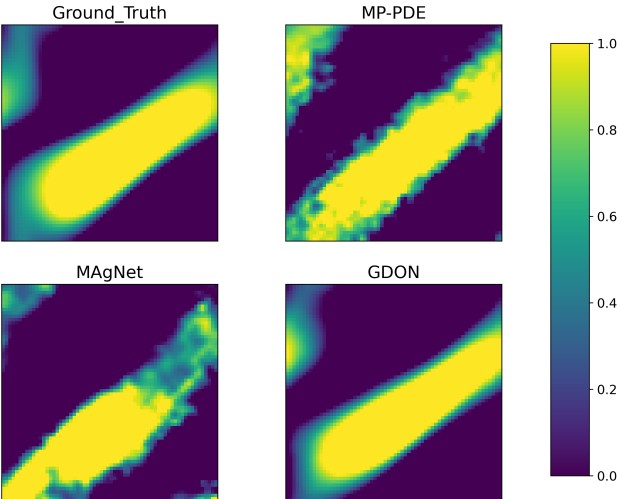

Figure 5: One snapshot for N-S equation data using MP-PDE, MAgNet, and GraphDeepONet.

### B.6 ENFORCING BOUNDARY CONDITION USING THE GRAPHDEEPONET

Utilizing the structure of DeepONet enables us to enforce the boundary condition $B[u] = 0$ as hard constraints. To elaborate further, we impose hard constraints for periodic boundary conditions and Dirichlet through a modified trunk net, which is one of the significant advantages of the DeepONet model structure, as also explained in [1]. For instance, in our paper, we specifically address enforcing periodic boundary conditions in the domain $\Omega$. To achieve this, we replace the network input $x$ in the trunk net with Fourier basis functions $\left(1, \cos(\frac{2\pi}{|\Omega|}x), \sin(\frac{2\pi}{|\Omega|}x), \cos(2\frac{2\pi}{|\Omega|}x), ...\right)$, naturally leading to a solution $u(t, \boldsymbol{x})$ ($\boldsymbol{x} \in \Omega$) that satisfies the $|\Omega|$-periodicity. As depicted in Figure 5, the results reveal that while other models fail to perfectly match the periodic boundary conditions, GraphDeepONet successfully aligns with the boundary conditions.

While our experiment primarily focuses on periodic boundary conditions, it is feasible to handle Dirichlet boundaries as well using ansatz extension as discussed in Choudhary et al. (2020); Horie & Mitsume (2022). If we aim to enforce the solution $\widetilde{u}(t, \boldsymbol{x}) = g(\boldsymbol{x})$ at $\boldsymbol{x} \in \partial\Omega$, we can construct the following solution:

$$\widetilde{u}(t, \boldsymbol{x}) = g(\boldsymbol{x}) + l(\boldsymbol{x}) \sum_{j=1}^{p} \nu_j[\boldsymbol{f}_{0:N-1}^1, \Delta t]\tau_j(\boldsymbol{x})$$

where $l(\boldsymbol{x})$ satisfies

$$\begin{cases} l(\boldsymbol{x}) = 0, & \boldsymbol{x} \in \partial\Omega, \\ l(\boldsymbol{x}) > 0, & \text{others.} \end{cases}$$

By constructing $g(\boldsymbol{x})$ and $l(\boldsymbol{x})$ appropriately, as described, we can effectively enforce Dirichlet boundary conditions as well. While the expressivity of the solution using neural networks may be somewhat reduced, there is a trade-off between enforcing boundaries and expressivity.

### B.7 EXPERIMENTS ON BURGERS' EQUATION

For Burgers' equation, we generate the uniform grid of 50 points in $[0, 16]$. We divided the time interval from 0 to 4 seconds uniformly to create 250 time steps. We started with 25 initial values for each segment, then predicted the values for the next 25 instances, and so on. The total number of

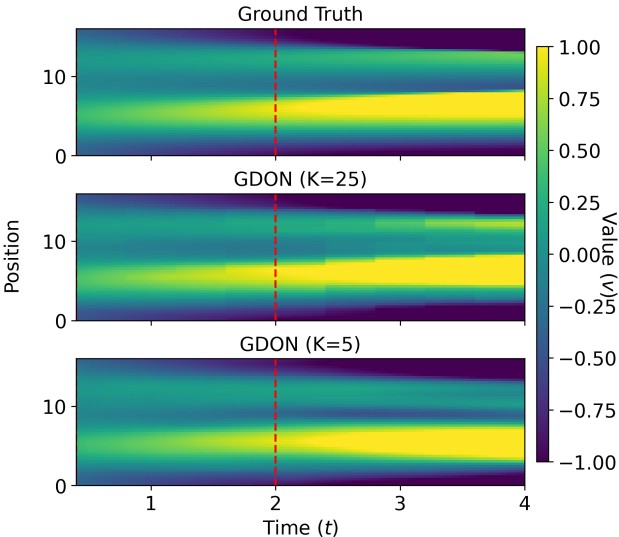

Figure 6: Comparison of solution profiles obtained from the Burgers' equation time extrapolation simulations using GraphDeepONet, with $K = 25$ and $K = 5$.

prediction steps is 9, calculated by dividing 225 by 25. In all experiments, we used a batch size of 16.

The training data consisted of 1896 samples, while both the validation and test samples contained 128 samples each. For irregular data, we selected 50 points from a uniform distribution over 100 uniform points within the range of 0 to 16 and made predictions on a fixed grid. The number of samples is the same as in the regular data scenario. To ensure a fair comparison in time extrapolation experiments, each model was assigned to learn the relative test error with a precision of 0.2 concerning the validation data. Our model conclusively shows superior extrapolation abilities compared to VIDON and DeepONet. Unlike DeepONet and VIDON, which tended to yield similar values throughout all locations after a given period, our model effectively predicted the local propagation of values.

### B.8 EXPERIMENTS ON 2D SHALLOW WATER EQUATION AND 2D N-S EQUATION

For the 2D shallow water equation, we generate the grid of $1024 = 32^2$ points for the regular setting. For irregular data, we selected an equal number of points from a uniform distribution over $128^2$ points within the rectangle $[-2.5, 2.5]^2$ and made predictions on a fixed grid. Figure 4 illustrates how we set up irregular sensor points for training GNN-based models and our model.

We evenly divided the time interval from 0 to 1 second uniformly to create 101 time steps. We started with 10 initial values for each segment, then predicted the values for the next 10 instances, and so on. The total number of prediction steps is 9, calculated by dividing 101-1=100 by 10. We remark that the values at $t = 1$ were excluded from the data set. In all experiments, we used a batch size of 4. For both regular data and irregular data, the training data consisted of 600 samples, while both the validation and test samples contained 200 samples each. Note that the MAgNet has the capability to interpolate values using the neural implicit neural representation technique. However, we did not utilize this technique when generating Figure 3, which assesses the interpolation ability for irregular data. For clarity, we've provided Figure 7 the predictions on the original irregular grid prior to interpolation.

In reference to the 2D Navier-Stokes equation, we apply the data from Li et al. (2020) with a viscosity of 0.001. For regular data, we generate the grid of $1024 = 32^2$ points. For irregular data, we selected an equal number of points from a uniform distribution over $64^2$ points within the rectangle $[0, 1]^2$ and made predictions on a fixed grid. We evenly divided the time interval from 1 to 50 seconds uniformly to create 50 time steps. We started with 10 initial values for each segment, then

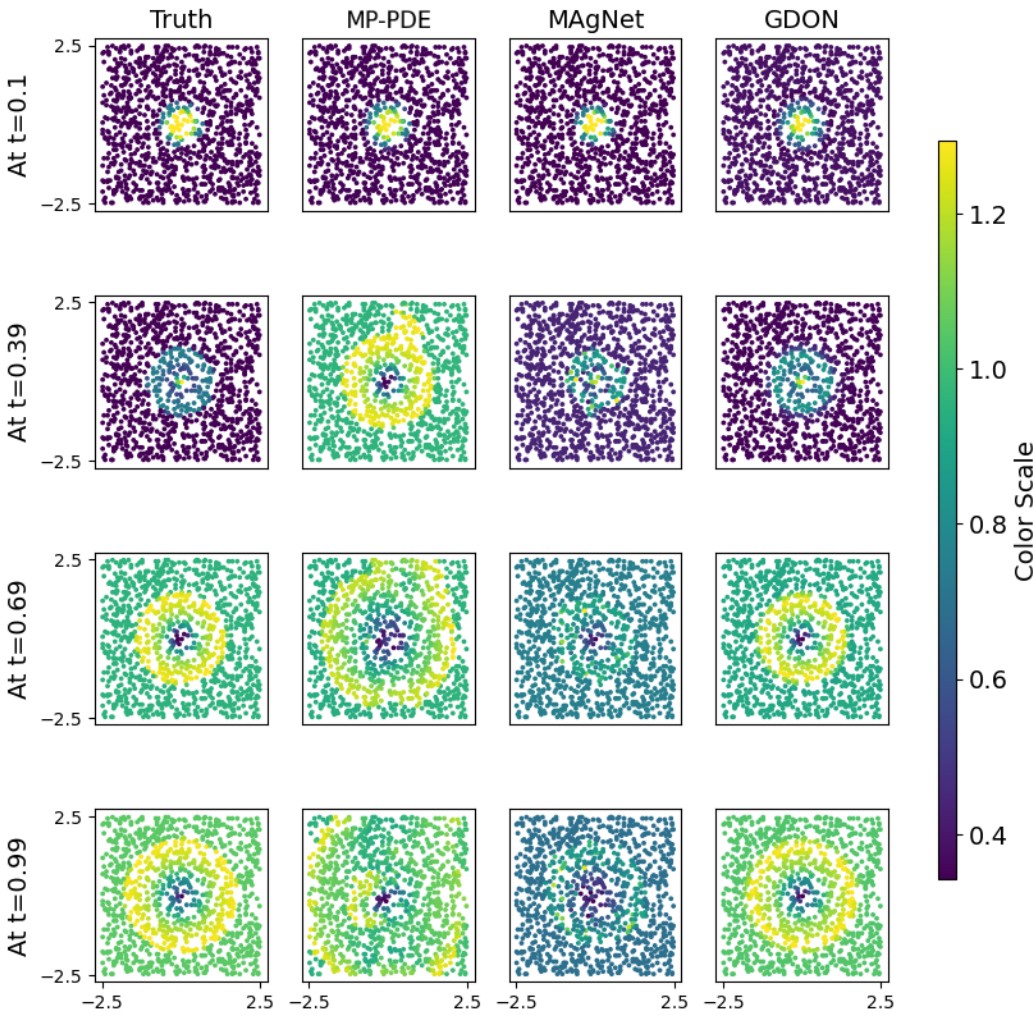

Figure 7: Ground truth solution and prediction profile for the 2D shallow water equation on a irregular grid.

predicted the values for the next 10 instances, and so on. The total number of prediction steps is 4, calculated by dividing 50-10=40 by 10. In all experiments, we used a batch size of 4. For both regular data and irregular data, the training data consisted of 600 samples, while both the validation and test samples contained 200 samples each.

