# OpenReview forum: "Learning time-dependent PDE via graph neural networks and deep operator network for robust accuracy on irregular grids"
_ICLR.cc/2024/Workshop/AI4DiffEqtnsInSci — AI4DiffEqtnsInSci @ ICLR 2024 Poster_

### Official Review · Reviewer_oNee · 2024-02-21
**Nice work, but some details on the relevance of the work are missing**

**Rating:** 7
**Confidence:** 4

**Review:**

in the introduction, the authors say "Message Passing Neural PDE Solver (MO-PDE) ... a limitation of their approach is that it can only predict the solution ... on the same grid used as input"

The authors should better clarify:
1. why this is a problem for practical simulation applications
2. hHow the GraphDeepONet proposed approach addresses and solves this limitation.

---

### Meta-Review · Area_Chair_GUNZ · 2024-02-29

**Recommendation:** Accept (Poster)

**Metareview:**

This paper presents a GNN method on surrogate models. The paper needs to clarify on the motivation, which can be address during the camera ready version.

---

### Decision · Program_Chairs · 2024-03-01

Accept (Poster)